# Alterations in Gene Expression and the Fatty Acid Profile Impact but Do Not Compromise the In Vitro Maturation of Zebrafish (*Danio rerio*) Stage III Ovarian Follicles after Cryopreservation

**DOI:** 10.3390/ani13223563

**Published:** 2023-11-18

**Authors:** Fernanda de Mello, Daniel Jaen Alonso, Natália Pires Vieira Morais de Faria, Victor Hugo Marques, Ethiene Fernandes de Oliveira, Paulo Henrique de Mello, Leandro César de Godoy, Renata Guimaraes Moreira

**Affiliations:** 1Department of Physiology, Bioscience Institute, University of Sao Paulo (IB/USP), 101 Matão Street, Travessa 14, Butantã District, São Paulo 05508-090, SP, Brazil; contato.dnalonso@gmail.com (D.J.A.); nataliapires96@gmail.com (N.P.V.M.d.F.); victor.marqueslorenti@gmail.com (V.H.M.); renata.fish@gmail.com (R.G.M.); 2Aquaculture Center, São Paulo State University “Julio de Mesquita Filho” (CAUNESP), Access Road Professor Paulo Donato Castellane, Vila Industrial District, Jaboticabal 14884-900, SP, Brazil; feroliver.thi@gmail.com; 3Beacon Development, King Abdullah University of Science and Technology, 2713, Jeddah 23955, Saudi Arabia; paulombio@gmail.com; 4Department of Animal Science, Federal University of Rio Grande do Sul, 7712 Bento Gonçalves Avenue, Agronomia District, Porto Alegre 91540-000, RS, Brazil; godoy@ufrgs.br

**Keywords:** vitrification, aquaculture, gamete quality, oogenesis, folliculogenesis, fertility preservation

## Abstract

**Simple Summary:**

This study discusses the use of vitrification, a technique for cryopreserving ovarian follicles, as a strategic tool for advancing aquaculture and species conservation. Vitrified gametes can be stored for extended periods, thus ensuring the availability of valuable genetic material for captive breeding and conservation programs. Alterations were found after vitrification, but the oocytes were still able to grow and mature, indicating that vitrification does not compromise in vitro maturation capability. This suggests that oocyte vitrification is a valuable technique for aquaculture purposes.

**Abstract:**

The vitrification of ovarian follicles is a strategic tool that may contribute to advances in aquaculture and the conservation of many important species. Despite the difficulties inherent to the cryopreservation of oocytes, some successful protocols have been developed for different species, but little is known about the capacity of oocytes to develop after thawing. Therefore, the profiles of the reproductive pathway genes and fatty acid membrane composition during the initial stages of development were analyzed in fresh ovarian follicles and follicles after the vitrification process. There were differences in the expression of the hypothalamic–pituitary–gonad axis genes during the follicular development in the control group as well as in the vitrified group. Similarly, alterations in the composition of fatty acids were observed after vitrification. Despite this, many alterations were observed in the vitrified group; more than half of the stage III ovarian follicles were able to grow and mature in vitro. Therefore, the vitrification of ovarian follicles may impact them at molecular and membrane levels, but it does not compromise their capability for in vitro maturation, which indicates that the technique can be a strategic tool for aquaculture.

## 1. Introduction

Cell cryopreservation is a unique tool that can be used in the aquaculture industry to preserve the genomes of domesticated species during the many steps of genetic selection. It can diffuse genetic progress, and it facilitates broodstock management by extending or delaying offspring production [1]. Vitrification is an effective cryopreservation technique in which ultra-rapid freezing transforms the cells from a liquid state to a glassy solid state without the formation of ice crystals [2]. This technique is cost-effective when compared to slow cooling [3,4]. Successful vitrification protocols for the semen of more than 200 fish species are available; however, for many other fish species, they are still being optimized. Notwithstanding, the vitrification of ovarian follicles is limited due to the characteristic of the vitellogenic oocyte, and it is widely acknowledged as being very difficult to perform in aquatic species [3,5,6]. Fish oocytes are more challenging to cryopreserve due to their larger size and large quantity of intracellular lipids, which can compromise the success of cryopreservation [6,7]. Nonetheless, vitrification has been successfully carried out in oocytes and embryos (blastocysts and two-cell-stage embryos) from mice by optimizing the process [8,9], in sheep embryos for embryo transfer programs under extensive conditions and using straws to perform the entire process [10,11], and in human oocytes, thus allowing increased pregnancy rates via artificial reproductive technology [12,13]. A number of studies have been conducted to develop optimized protocols that enable fish oocyte cryopreservation during the early stages of development [14,15,16,17]. Despite these optimized protocols, in vitro studies that test the viability and capacity for oocyte growth and maturation after thawing are scarce [17].

The resumption of oocyte meiosis is a critical step for the progression of its development, which requires a close collaboration of a variety of hormones and growth factors. As far as we know, the development of the ovarian follicle in fish, as well as in all vertebrates, occurs through a complex network that involves different neuropeptides, neurotransmitters, and pituitary hormones through the hypothalamus–pituitary–gonad (H-P-G) axis [18,19,20]. The development of the ovarian follicle is marked by the expression of many genes, some of which are then translated into proteins, while others will be expressed and accumulated to be used by the embryo until the maternal–zygotic transition. The translation of some of these genes is recognized as essential for the progression of oocyte maturation and development [21]. Not only gene expression but also all the signaling of the H-P-G axis during oogenesis, as well as cell membrane conformation, are essential for the growth and maturation of ovarian follicles, allowing for fertilization and offspring generation [17,18,22,23]. Fatty acids (FAs) maintain the cellular membrane conformation and can be stored in the gonad for structural purposes or catabolized for energy during oogenesis [24,25]. Moreover, specific FAs are metabolized for eicosanoid synthesis, which in turn stimulates steroid synthesis in the ovary, triggers oocyte maturation, and affects the sexual behavior of females [7]. Temperature is a critical factor for the membrane as it affects the degree of saturation of fatty acids (FAs), thereby determining membrane fluidity [26,27]. Despite this, the dynamic changes in gene expression and fatty acid pattern to which the follicles need to adjust to remain viable post-thawing are not entirely clear in teleosts.

The understanding of how the cell is affected by vitrification and how it adjusts to keep growing and maturing post-thawing is pivotal for the development of novel approaches for reproductive management in fish farming and can contribute to the conservation of endangered species. Therefore, the main objective of the present study was to evaluate alterations in gene expression and the fatty acid profile using an established vitrification protocol for stage III ovarian follicles in zebrafish. Subsequently, we performed an in vitro culture of the vitrified and thawed follicles to assess whether the process could compromise their development capacity.

## 2. Materials and Methods

### 2.1. Fish Treatment and Ovarian Follicle Collection

Ovarian follicles were obtained from female zebrafish with fully grown ovaries after the fish were euthanized with a lethal dose of tricaine (0.6 mg/mL). Ovaries were immediately removed after decapitation and were placed in a Petri dish containing a 90% Leibovitz L-15 medium (pH 9.0) (Sigma-Aldrich, St. Louis, MO, USA). Ovarian tissue fragments were collected manually using forceps and fine needles under a dissecting stereomicroscope (Leica EZ4). The follicles were classified according to their developmental stage: primary growth (PG) or stage I (less than 0.1 mm in diameter); previtellogenesis (PV) or stage II (~0.30 mm in diameter); and vitellogenesis (VG) or stage III (~0.45 mm in diameter) [28]. Developmental stages were separated into two groups for gene expression analysis. The in vitro maturation experiment and fatty acid profile measurements were performed only with VG follicles in both the control (fresh) and the vitrified groups.

### 2.2. Vitrification of Ovarian Follicles

The follicles that were to be vitrified were separated soon after the stages of development were identified after follicle isolation, as described above. For vitrification, first, all follicles were exposed to an equilibrium solution (ES), which was composed of L-15 medium + 1.5 M methanol + 2.25 M dimethyl sulfoxide (DMSO) + 0.25 M sucrose, for 15 min at room temperature. Subsequently, the follicles were exposed to a vitrification solution (VS), which was composed of L-15 medium + 1.5 M methanol + 5.5 M DMSO + 0.5 M sucrose, for 15 min [15]. Afterward, the vitrified follicles were stored in 0.5 mL cryotubes, plunged directly into liquid nitrogen, and remained stored until analysis.

### 2.3. Total RNA Extraction and Quantitative RT-PCR

Ovarian follicles were lysed with a Precellys Evolution Homogenizer (Ozyme, Bertin Technologies, Montigny-le-Bretonneux, France) in TRI Reagent (TRI^®^ Sigma-Aldrich, St. Louis, MO, USA) for total RNA extraction, which was carried out according to the manufacturer’s recommendations. For the expression analysis of selected genes (Table 1), 1 µg of total RNA was reverse-transcribed (RT) using a High-Capacity cDNA Reverse Transcription Kit (4368814, Applied Biosystems, Foster City, CA, USA). The cDNA was diluted (1:20), and the reaction was carried out using qRT-PCR with 5 μL of diluted cDNA, 10 μL of Power SyberGreen PCR Master Mix (Thermofisher, Waltham, MA, USA), and 6 μM of each primer, to a total volume of 20 μL. Primer efficiencies (in between 85% and 100%) were calculated using serial cDNA dilutions (1:1, 1:4, 1:16, 1:64, and 1:256) from a pool of all the samples, in triplicate, and amplification was assessed in a negative control without cDNA. The amplification conditions were optimized before the analysis of expression. qRT-PCR was conducted using the StepOne Plus system (Applied Biosystems, Foster City, CA, USA) with the following conditions: 95 °C for 2 min; 40 cycles of 95 °C for 15 s and 60 °C for 1 min. Standard curves were generated using five serial cDNA dilutions (from 1:1 to 1:256) from a pool of all the samples. The relative abundance of target cDNA was calculated from a standard curve using Applied Biosystem StepOne V.2.0 software (Applied Biosystems, Foster City, CA, USA). Relative expression data were derived from three biological replicates and duplicate technical replicates, and the data were normalized using the geometric mean of the expression of the β-actin and ef1α genes. Oocyte growth and maturation genes were analyzed according to the follicle’s developmental stage, comparing the same stage in the vitrified group and in the control group.

### 2.4. Fatty Acid Profile

For the analysis of the FA profile, we used stage III ovarian follicles collected from different females, which were kept for eight weeks in the laboratory and fed four times a day with TetraMin^®^ dried flakes (Tetra, Melle, Germany). Females were maintained in aerated and temperature-regulated (27 °C) water in 40 L tanks under a light–dark photoperiod of 14/10 h. Then, total lipids from stage III ovarian follicles, from the fresh and vitrified groups, were extracted using the conventional method [29], and the lipid extract was separated into the phospholipid (PL) and triglyceride fraction (TG) [30]. Subsequently, the lipid extracts of the different fractions were methylated using acetyl chloride (5% HCl in methanol) [31]. Fatty acids were analyzed as methyl esters using a gas chromatograph (Scion, 436) equipped with a flame ionizer (FID) and an auto-injector (Scion 436-GC, 8400 Autosampler, LC Nieuwegein, Nederland). The capillary column used to analyze the fatty acids was a CP Wax 52 CB, and the following temperature program was used: 170 °C for 1 min, followed by a ramp rate of 2.5 °C/min until reaching 240 °C, and a final waiting time of 5 min, totaling a 31 min run time. The temperature was 250 °C in the injector and 260 °C in the FID. A methyl ester standard (FAME) (Supelco, 37 comp., Louis, MO, USA) was used to identify fatty acids based on retention time. Data are presented as means and standard error of the mean.

### 2.5. In Vitro Culture of Fresh and Vitrified Stage III Ovarian Follicles

The in vitro maturation of ovarian follicles was examined as described previously [32]. Ovarian follicles were manually isolated from the ovaries with a fine needle and gentle pipetting, and then, stage III follicles were identified as described above. Ten follicles per well were placed (in 12-well culture plates) containing 4 mL of the Leibovitz medium (L-15, 90%), supplemented with 100 µg of gentamicin and 0.8 µg/mL 17α-hydroxy-20β-dihydroxyprogesterone, 4-pregnene-17α-20β-diol-3-one; C_21_H_32_O_3_ (Cayman 16146); and the maturation-inducing hormone (DPH). The control group was composed of follicles from different females, collected on the day of culture in L-15 medium. The vitrified group included follicles that had previously been cryopreserved, as described above. The vitrified group was warmed by placing the cryovials in a water bath for 30 s at 28 °C, and subsequently, they were exposed to three thawing solutions at 22 °C: the first containing 1 M sucrose for 1 min, followed by a second solution containing 0.5 M sucrose for 3 min and, finally, a third solution containing 0.25 M sucrose for 5 min. The samples were washed three times in the L-15 medium and assessed under a magnifying glass to verify post-thawing viability. Dead follicles (due to warming) were eliminated, and the groups formed of only viable follicles were distributed (10 follicles per well) in a 12-well plate. The plate containing the follicles was kept at 27 °C for 270 min; the viability of the follicles and their maturation capacity were assessed every hour until the follicles reached stage V development, in other words, until they were mature. A follicle was considered mature when the disappearance of the germinal vesicle (GBVD) and maximum transparency of the oocyte was observed since the non-mature follicles contain intact germinal vesicles and remain opaque [28].

### 2.6. Ovarian Follicle Viability Assay

After the isolation of each stage III follicle, a portion of the follicles was separated in order to perform a membrane integrity test using trypan blue (TB). In both the control and vitrified groups, a portion of the isolated follicles was incubated in 0.4% TB for 3 min. Then, the follicles were observed under a microscope, and those that were not stained blue were considered to have an intact membrane, while those stained blue were considered to have a damaged membrane, i.e., non-viable [15]. Only isolations with more than 95% viable follicles were used in the analysis.

Follicle viability was estimated after in vitro maturation by employing the rhodamine 123 fluorescent probe in combination with propidium iodide (PI). The former penetrates living follicles and emits a bright green fluorescence, whereas PI penetrates only cells that have lost membrane integrity and emits in the red fluorescence range. Aliquots of the follicle suspension (120 oocytes/plate) were stained by adding 100 µM rhodamine 123 and incubated for 15 min in the dark at 22 ± 1 °C. Following this, 100 μM of PI was added, and the samples were further incubated for 15 min at 22 ± 1 °C in the dark. To validate the rhodamine 123 and PI staining results, follicles were killed by incubating for 1 h in glutaraldehyde 4% in the L-15 medium.

### 2.7. Statistical Analysis

Oocyte growth and maturation genes were analyzed according to the follicle development stages, comparing the same stage in the vitrified and control groups. The in vitro maturation experiment and fatty acid profile were analyzed by comparing the control group and the vitrified group. The analysis was carried out using the non-parametric Wilcoxon/Kruskal–Wallis test and one-way ANOVA, and the Tukey test (*p* < 0.05) was used for post hoc analysis. Gene expression data were normalized using the geometric mean of the β-actin and ef1α genes, with the results presented as mean ± standard error. All analyses were conducted using the R software (RStudio RStudio: Integrated Development for R. RStudio, version: 2023.09.1+494, PBC, Boston, MA, PBC, Boston, MA, USA).

## 3. Results

### 3.1. Reproductive Pathway Genes

In this study, some of the main genes involved in oocyte growth and maturation were analyzed, as well as two apoptosis markers (Figure 1). In the control group, *fshr* and *lhcgr* mRNA levels showed differences throughout oogenesis. The highest *fshr* mRNA level was found in follicles during the previtellogenesis (PV) stage, which was 12.5-fold higher than in follicles in the primary growth (PG) stage and 6-fold higher than in follicles in the vitellogenesis (VG) stage (*p* = 0.02). In contrast, the highest level of *lhcgr* mRNA was observed in VG follicles, increasing its expression 50-fold when compared to PV follicles and 12-fold when compared to PG follicles (*p* = 0.01). Similarly, in the control group, there were also differences in *IGF-I* and *cyp19a1* mRNA levels, in which IGF-I had the highest level of mRNA in primary growth (PG) follicles when compared with PV (2-fold) and VG (1.25-fold) stages; and *cyp19a1* had its highest level in PV follicles, which was 3-fold higher than PG follicles and 1.5-fold higher than VG follicles. After vitrification, changes were observed in mRNA levels of both genes (*p* < 0.05). Notably, *cyp19a1* expression increased in the vitrified ovarian follicle during the PV and VG stages. Moreover, the *IGF-I* mRNA level increased after vitrification, but this increase was only observed during the PG and PV stages. Nonetheless, the *lhcgr* mRNA level was lower after vitrification in PG and VG (*p* = 0.001) stages, and there was no difference observed in the *fshr* mRNA level after vitrification.

Steroidogenic acute regulatory protein (*Star*) and estrogen receptor (*ESR*) genes showed the highest mRNA levels in the control group (*p* < 0.0001), in the PV stage for *star* (*p* = 0.043) and *esr2a* (*p* = 0.005) genes. After vitrification, there was a decrease in mRNA levels; there were no differences in the expression of these genes between the PG, PV, and VG stages of development when compared to the control group (*p* = 0.3). In the same way, the cell apoptosis markers genes *casp3* (*p* = 0.004) and *tp53* (*p* = 0.01) exhibited differences in the PG stage, in which the highest mRNA levels were observed in the control group.

### 3.2. Fatty Acid Profile

In the TG and PL fractions, the percentages of total saturated fatty acids (SFAs) were higher in the control group (*p* < 0.05). In the PL fraction, the control group showed a significantly higher percentage in C14:0 (*p* = 0.021) (Figure 2A,B). In the TG fraction, monounsaturated fatty acids (MUFAs) and oleic acid C18:1n-9 showed a higher percentage in the vitrified group (*p* = 0.002). These percentages were directly reflected in the total MUFAs, which presented 44.07 ± 1.49% in the vitrified group, while the control group presented 31.25 ± 2.37% (Figure 2C,D). Arachidonic acid (ARA) C20:4n-6 showed significantly higher levels (*p* > 0.05) in the vitrified group in both of the analyzed fractions. The same occurred with docosahexaenoic acid (DHA) C22:6n-3 (*p* > 0.05). In the PL fraction, linoleic acid C18:2n-6 increased by 59.86 ± 0.18% after vitrification (see Appendix A) (*p* = 0.0031), consequently resulting in higher percentages of polyunsaturated fatty acids (PUFAs) within the same fraction. In the same way, the high levels of DHA directly affected the total sum of long-chain PUFAs (LC-PUFAs), which consequently showed higher percentages in the vitrified group (*p* = 0.001) and was approximately 16% higher than in the control group (Figure 2E,F). In contrast, there was a decrease in the MUFA vaccenic acid C18:1n-7 after vitrification (34 ± 0.15%) (*p* < 0.05). The fatty acid profile of the neutral and polar fractions are presented as Appendix A (Appendix A).

### 3.3. In Vitro Maturation of Fresh and Vitrified Follicles

For the in vitro culture, vitrified follicles were thawed, and tests using trypan blue were carried out to detect inviable cells. After 270 min, differences were observed in the percentage of mature follicles in the control (86 ± 1.29%) and vitrified (52 ± 1.52%) groups (*p* = 0.05) (Figure 3). On the other hand, there was no difference in the number of immature follicles after the culture period for both of the groups (*p* = 0.06). The percentage of mature follicles refers to the number of follicles in which the disappearance of the germinal vesicle (GVBD) and maximum transparency of the oocyte were observed (Figure 4). The presence of a micropyle located at the animal pole of the cells that matured at the end of the culture period was observed; however, this was not used as a criterion to classify the follicles.

A viability assay using rhodamine 123 confirmed that, after the culture period, the cells that matured were physiologically functional (Figure 5). Therefore, a difference was observed in the number of damaged follicles between the control (8.3 ± 1.32%) and vitrified (60.3 ± 2.09%) groups (*p* = 0.03) (Figure 3). Cells were considered functionally damaged if they suffered death during the process, unlike immature follicles that remained alive until the end of the culture period.

## 4. Discussion

### 4.1. Impact of Follicle Vitrification on the Expression of Reproductive Pathway Genes

Important genes that drive the development of ovarian follicles in different fish species and humans [21,33,34,35] were quantified. The results show that gonadotropin receptors (*fshr* and *lhcgr*), *IGF-I*, *star*, *Cyp19a*, and estrogen receptors (*esr1*, *esr2a*, and *esr2b*) are dynamically expressed during oogenesis in fresh zebrafish follicles (control group), which is consistent with what has been described for other fish species [36,37]. Indeed, significant alterations were observed in mRNA levels following the vitrification process, which can affect the dynamic expression necessary for follicular development. The highest level of mRNA for the *star* (steroidogenic acute regulatory protein) gene was observed in the control group; however, contrary to expectations, no difference was detected between the analyzed phases. In teleosts, as in other non-mammalian vertebrates, the *star* gene is expressed in theca cells, where it is responsible for transporting cholesterol into the mitochondria to produce testosterone. In follicular cells, this testosterone will be converted into estradiol-17β (E2) by enzymes such as *cyp19a1*, the ovarian form of cytochrome P450 aromatase (*oP450arom*, *P450c19a1*) [36]. Therefore, the highest expression of the star gene was expected during the PG stage since this gene provides the substrate for the production of testosterone, which is the initial step for androgen production. The *cyp19a1* gene also plays a crucial role in oocyte growth (vitellogenesis) by converting testosterone into E2. Nonetheless, *star* expression decreased sharply after vitrification, contrary to what was observed in the *cyp19a1* expression, which was markedly enhanced through vitrification. This indicates changes in steroid biosynthesis. Despite being stimulated by FSH in the follicular cells of the PV stage, *cyp19a1* subsequently decreased as a consequence of the surge in LH [38]. The expression of *oP450arom* (*cyp19a1*) declines sharply in follicles immediately before oocyte maturation and becomes undetectable in post-vitellogenic follicles during the meiotic maturation stage [32,33]. In the present study, both the PV and VG stages were found to have an increase in *cyp19a1* mRNA levels after vitrification. This result can be partly explained by the decrease in LH receptors (*lhcgr*) observed in VG follicles after vitrification. These findings in the ovarian follicles of the control group are in accordance with previous reports [39,40]; however, more importantly, the present study revealed the deregulation of *lhcgr* and *cyp19a1* after vitrification. Reproduction cannot occur without the proper functioning of *lhcgr*, and it would most likely not be possible to develop viable zebrafish embryos as a result of these gene expression changes [41]. FSH-deficient zebrafish (*fshb*^−/−^) have been shown to have significantly delayed ovary development, although both sexes can be fertile. In contrast, LH-deficient zebrafish (*lhb*^−/−^) show normal gonadal growth, but the females fail to spawn [23]. In mice, LH-null (*lhb*^−/−^) mice are viable but demonstrate post-natal defects in gonadal growth and defects in folliculogenesis and function, which results in infertility [42].

The high expression of estrogen receptors (*esr1*, *esr2a*, and *esr2b*) in the PG follicles of the control group may be due to the increase in E2 synthesis. This hypothesis is reinforced by the high expression of *esr2a*, which is recognized as the receptor with greater sensitivity to E2; *esr2a* was expressed 30-fold more during PG than during the PV stage (although no statistical difference was observed). After vitrification, there was a decrease in *esr2a* and *esr2b*. Furthermore, estrogen receptors can bind to growth factors, such as *IGF-I*, in the non-classical pathway (non-genomic action) [43,44,45], which would explain the decrease in *IGF-I* expression during the VG stage after vitrification. In the same way, *IGF-I* gene expression was altered through vitrification, showing a greatly increased expression during the PG and PV stages. *IGF-I* has been implicated in the transition of PV follicles, and changes in its expression could have implications for follicle growth [46]. In addition, *IGF-I* increases the responsiveness of the follicle to 17α,20β-dyhydroxy-4-pregnen-3-one, suggesting a role in promoting maturational competence in vitro [47].

On the whole, these data suggest that the *IGF-I* and *cyp19a1* gene expressions increased due to a decrease in *lhcgr* expression after vitrification. The *IGF-1* receptors present in the outer layer of the ovarian follicles aim to facilitate an increase in the synthesis of LH and FSH and also stimulate the activity of *cyp19a1* [46]. This modification in expression dynamics may be a strategy for increasing LH uptake by increasing *lhcgr* expression during the VG stage.

In the context of in vitro development, the changes observed in the expression of *cyp19a1* and *lhcgr* were not critical. This is because the production and release of E2 would not stimulate the production of vitellogenin by the liver, nor would it inhibit the synthesis of FSH or stimulate the synthesis of LH in the hypothalamus–pituitary–gonad axis, since the follicle is isolated. In this experimental approach, oocyte maturation was stimulated with the addition of the maturation-inducing hormone (DPH) to the culture medium, which promotes the resumption of meiosis. This can be observed through the breakdown of the germinal vesicle and the emergence of the micropyle. Nevertheless, we cannot estimate whether the embryo generated from a follicle with such changes in gene expression would develop normally since it would develop without having completed total vitellogenesis.

The reducing gene expression of the apoptosis markers *TP53* and *Casp3* in PG oocytes after vitrification may be a consequence of the low temperatures in the maintenance of normal tissue function after thawing. Apoptosis is a natural process that helps maintain tissue homeostasis by eliminating damaged or unwanted cells. However, the vitrification process can potentially lead to tissue damage. Reducing apoptosis markers can play a role in promoting cell survival and maintaining tissue integrity, thereby facilitating oocyte restructuration after thawing. In atretic ovaries of *Merluccius merluccius*, apoptosis genes *TP53* and *casp3* are upregulated together with autophagocytosis marker genes, and vitellogenesis is the process in which both apoptosis and autophagy genes exhibit their highest expression [48]. Nevertheless, the same higher expression of these markers was not detected in the other stages of follicular development in the control group. In the same way, studies in mammals, including humans, found no differences in apoptosis-related gene expression after the vitrification of oocytes and thawing.

### 4.2. Impact of Vitrification on the Fatty Acid (FA) Profile

The control group exhibited higher percentages of SFAs. Studies in mammals have demonstrated that the oxidation of FAs is an energy provider in oocytes, and the metabolites of some lipids are also regulators of oocyte meiosis and maturation [49]. The elevated percentage of SFAs in the control group, particularly in the TG, could be strongly associated with these cellular processes, as in mammals, since SFAs serve as a primary source for energy generation through the breakdown of triglycerides and shorter-chain FAs. Less complex TG, such as C12:0 or C14:0, is known to be physiologically less important and present in several species’ follicular fluids. Further investigations in teleost fish are required in order to comprehend their role in energy production for ovarian cells and oocytes. One in vitro study suggested that, as oocytes mature, lipolysis becomes more active [49]. Based on the obtained data, the disparity between the control group (which displayed a higher SFA percentage than the vitrified group) could be attributed to greater energy expenditure in the ovarian follicles of the females in stage III in the vitrified group.

The presence of 30–40% oleic acid 18:1n-9 in stage III vitrified follicles may suggest accumulation, possibly for some metabolic purpose, which could be advantageous. Oleic acid is directly associated with the number of retrieved oocytes and embryo formation after fertilization, but the high content of 18:1n-9 in the follicular fluid has been shown to negatively impact oocyte development in mammals [49]. Nonetheless, it is not clear how the changes observed in the fatty acid profile after vitrification can impact the oocyte’s ability to be fertilized and generate viable individuals. Like other vertebrates, freshwater fish lack the enzymes to synthesize unsaturated FAs such as linoleic acid 18:2n-6. Therefore, they are considered essential FAs and must be provided in the diet to avoid nutritional deficiencies [50]. On the other hand, vitrification expressively increased the levels of linoleic acid in the PL fraction, consequently resulting in higher percentages of PUFAs within the same fraction of this group. Among ectothermic vertebrates, one typical response to temperature changes involves membrane restructuring, which can be facilitated by specialized proteins embedded in the lipid bilayer [50]. These proteins transmit signals to the intracellular environment and initiate the necessary adjustments, potentially explaining the increase in PUFAs and LC-PUFAs in the PL fraction.

The presence of high percentages of LC-PUFAs, specifically docosahexaenoic acid (DHA, C22:6n-3) and arachidonic acid (ARA, C20:4n-6), in both analyzed fractions in the vitrified group is promising, as these FAs play essential roles in the physiology and development of teleosts. ARA originates from the conversion of linoleic acid (C18:2n-6) through a pathway involving a series of enzyme elongation and desaturation steps; together with eicosapentaenoic acid (EPA, C20:5n-3) and DHA, these LC-PUFAs are necessary for normal fish development, including growth and reproduction. ARA is a primary precursor of prostaglandins and is a crucial mediator in steroidogenesis, oocyte maturation, and ovulation [50,51,52].

Vitellogenin is a fundamental carrier of lipids for oocytes in fish species, with vitellogenic teleosts containing approximately 20% of lipids, up to 80% of which can be PL, which is rich in PUFAs and LC-PUFAs [52]. In mammals, PUFAs may comprise about 20% of the total FAs in bovine oocytes and embryos, and this lipid accumulation is an energy reserve for early embryonic development [7,53]. In oviparous teleosts, in which the offspring develops without the direct supply of nutrients from the mother, the yolk plays an even more crucial role [49,54]. DHA is an essential FA for the development of nervous tissues and contributes to the fluidity of biological membranes [24,55], which are crucial for developing a future embryo. Therefore, the presence of ARA and DHA in higher quantities in the vitrified group may be promising for future studies.

Previous studies by our group showed that vitrification altered the overall FA profile in ovarian follicles in different stages of development [56]. There was an increase in SFAs and a reduction in PUFAs, potentially leading to a more rigid and less fluid cellular membrane. For stage III ovarian follicles, both TG and PL had a higher percentage of SFAs and a lower percentage of PUFAs, possibly caused by vitrification. These results suggest that the membranes of vitrified ovarian follicles are less fluid, although this did not compromise the composition of the FAs.

### 4.3. Impact of Vitrification on In Vitro Development Capacity

To infer whether the alterations detected in gene expression and the fatty acid profile after vitrification can compromise the oocyte’s capacity for development, we carried out an in vitro culture of the thawed follicles, using fresh follicles as a control. We successfully reproduced the protocol described for zebrafish species [32] and achieved 86 ± 1.29% of mature follicles at the end of the culture period. The high percentage of follicles with intact membranes obtained to perform the in vitro culture was essential for the accuracy of the data to compare the control and vitrified groups, reinforcing the efficiency of the care taken during the collection of the ovaries.

In this study, differences were observed in the percentage of maturation between fresh (86 ± 1.29%) and vitrified (52 ± 1.52%) follicles. Initially, the performance of freshly collected follicles deserves attention; however, obtaining over half of the maturation using follicles that have undergone vitrification demonstrates the success of cryopreservation despite all the impacts on cell physiology. Here, the follicles were considered mature in vitro when, at the end of 270 min, they showed maximum oocyte transparency, an event that is associated with yolk proteolysis during ovulation. In addition, GBVD, which is the nuclear migration toward the micropyle was also observed, an event corroborated by micropyle formation in the follicles of both fresh and vitrified groups. The micropyle is a small opening located at the animal pole of the cell, through which the sperm enters the mature oocyte [57]. Micropyle formation indicates that the cell is physiologically active, resuming meiosis I, which will culminate in the expulsion of the first polar body, although meiosis is completed only with fertilization. The physiologic cell activity was proven by the visualization of fluorescence detected in follicle mitochondria due to the active transport of ions across the membrane.

A higher number of dead cells from the vitrified group during the induction of maturation was observed, which may indicate the fragility of the cells due to the cryopreservation process. The ovarian follicles that matured in vitro after thawing showed morphological differences in terms of maturation progress and color in the fresh follicles. This is evidenced in studies using cryopreserved stage III ovarian follicles [2], which began to become semi-translucent and swollen thirty minutes after thawing, thus indicating changes in the structure of the yolk. Therefore, there is a possibility that vitrification damages the internal compartments of ovarian follicles via the release of proteases or the alteration of ionic transport mechanisms that can change the structure of yolk proteins [15]. The high number of dead follicles in the vitrified group during the culture period may be due to the toxicity of cryoprotective solutions. It is known that methanol and ethanol reduce the integrity of ovarian follicles in stages I and II of development [54]. Our results suggest that the combination of methanol and DMSO at the concentrations used in this study also caused damage to stage III follicles.

## 5. Conclusions

There was a decrease in the ability of the follicle to develop after vitrification. Although we used a well-established vitrification protocol for stage III zebrafish ovarian follicles, remarkable alterations were observed in the expression of genes involved in oocyte growth and maturation, and in the profile of the fatty acids that form the cell membrane after vitrification when compared to fresh follicles.

So far, the changes in the fatty acid profile seem to have the most impact on the cell; all vitrified–thawed follicles exhibited a fatty acid profile that suggests a more fragile membrane, including those that successfully matured in vitro. Notwithstanding, modifications were detected in the expression of genes that have a pivotal role in follicular development. However, judging by the rates of in vitro matured oocytes, it is evident that these alterations did not impede the oocyte’s capacity to thrive and mature. Over 60% of the vitrified–thawed oocytes matured in vitro, suggesting that the cells can rearrange and develop to reach maturation. In this way, the vitrification of ovarian follicles proves to be a valuable technique in aquaculture and can facilitate the preservation of genetic material of threatened species in germplasm cryobanks. Future studies are warranted to uncover and elucidate aspects of reproduction, especially those related to the mechanisms of fertilization and embryonic development, which go beyond the parameters evaluated in this study but for which our study is the basis. These investigations could contribute to a more comprehensive understanding of the post-cryopreservation oocytes’ physiology and, ultimately, advance our knowledge in this field.

## Figures and Tables

**Figure 1 animals-13-03563-f001:**
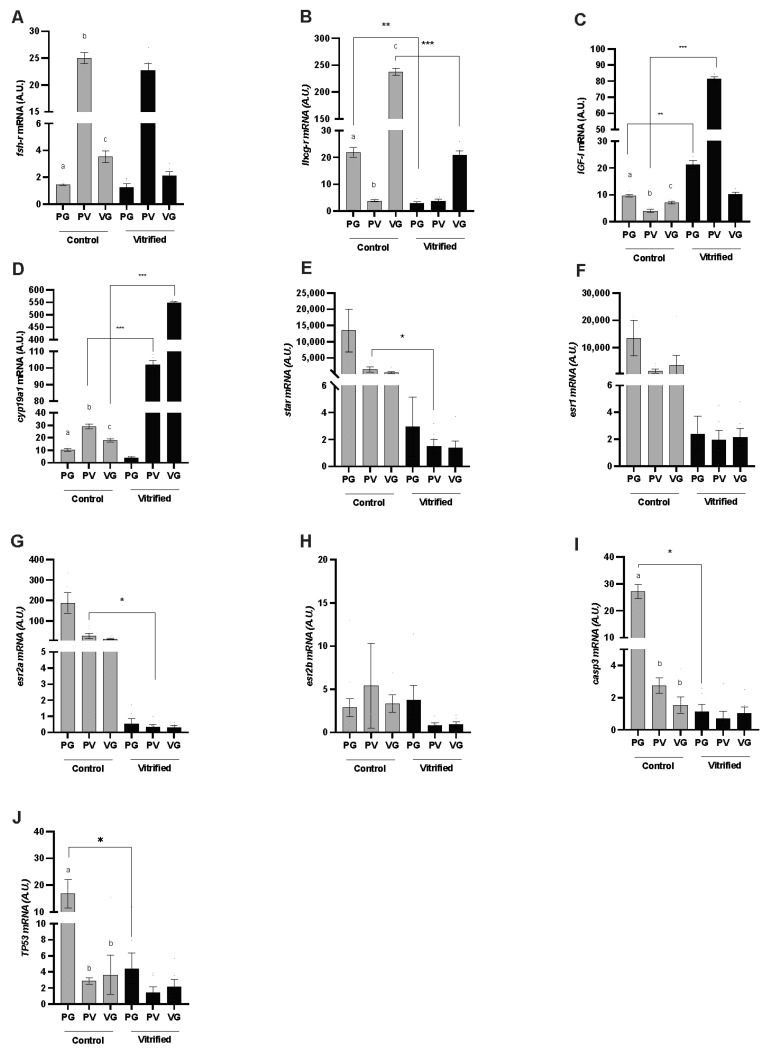
Expression profiles of transcripts for (**A**) follicle-stimulating hormone receptor (*fshr*), (**B**) luteinizing hormone/choriogonadotropin receptor (*lhcgr*), (**C**) insulin-like growth factor I (*IGF-I*), (**D**) cytochrome P450 family 19 subfamily A member 1 (*Cyp19a1*), (**E**) steroidogenic acute regulatory protein (*star*), (**F**) estrogen receptor 1 (*esr1*), (**G**) estrogen receptor 2a (*esr2a*), (**H**) estrogen receptor 2b (*esr2b*), (**I**) caspase 3 (*casp3*), and (**J**) tumoral protein 53 (*TP53*) genes in the ovarian follicles of zebrafish (*Danio rerio*) during primary growth (PG) (stage I), the previtellogenesis phase (PV) (stage II or cortical alveolus stage), and the vitellogenesis phase (stage III; VG), analyzed per group: control (fresh) (gray columns) and vitrified (black columns). The data are presented as mean and standard error (n = 5; each mean is from three biological replicates and duplicate technical replicates). Different letters indicate a difference between the developmental stages within the control group. Asterisks indicate differences between the stage of development in the vitrified versus the control group; * *p* < 0.05, ** *p* < 0.01, *** *p* < 0.001.

**Figure 2 animals-13-03563-f002:**
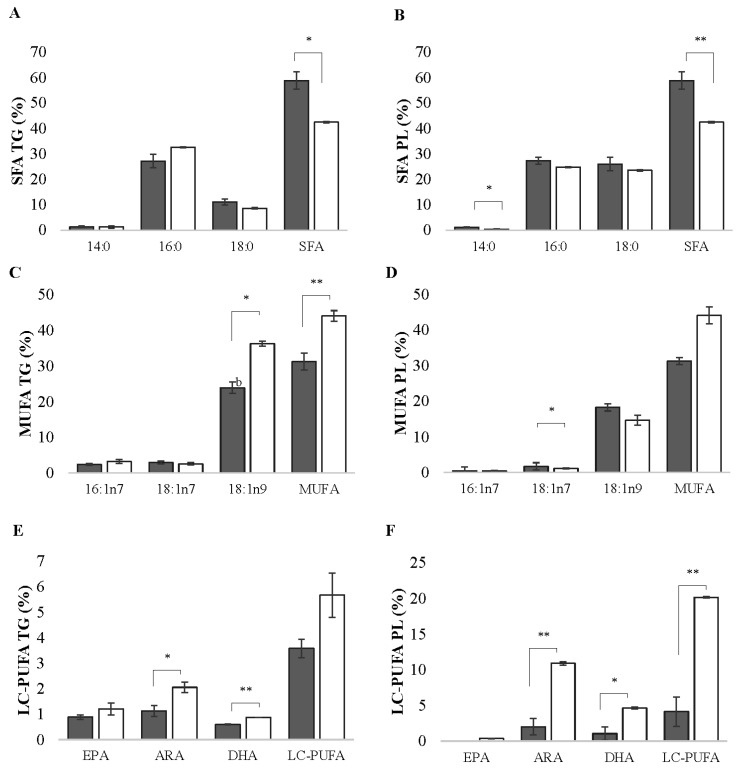
Fatty acid profiles of ovarian follicles in the previtellogenesis phase (stage III) of zebrafish (*Danio rerio*) for the control (grey column) and cryopreserved (white column) samples in the neutral (TG = triglycerides) and polar (PL = phospholipids) fractions for (**A**,**B**) saturated fatty acids (SFAs), (**C**,**D**) monounsaturated fatty acids (MUFAs), and (**E**,**F**) long-chain polyunsaturated fatty acids (LC-PUFAs). The data are presented as mean and standard error (n = 6; each mean is from three biological replicates and three technical replicates). Asterisks indicate differences between the stage of development in the vitrified versus the control group; * *p* < 0.05, ** *p* < 0.01.

**Figure 3 animals-13-03563-f003:**
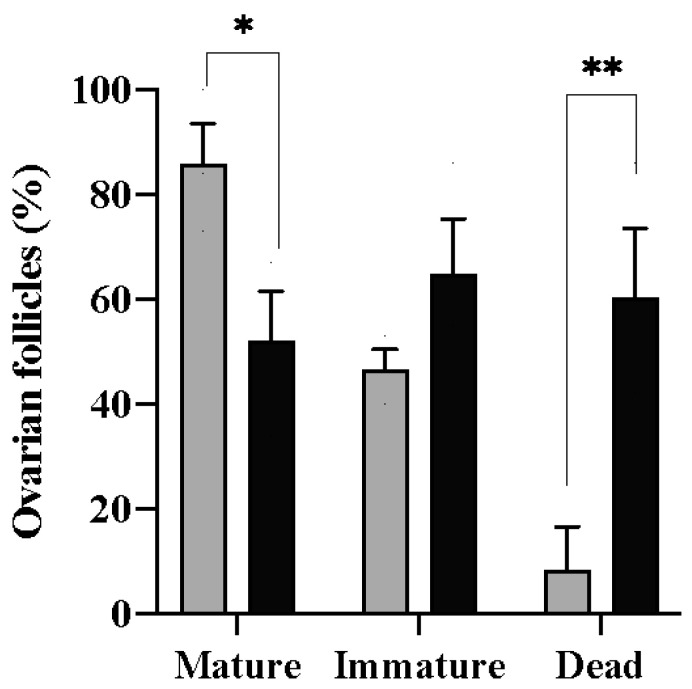
Percentage of zebrafish (*Danio rerio*) ovarian follicles that developed (mature), remained in stage III (immature), or did not survive (dead) after being cultured for 270 min in vitro at 27 °C. The data are presented as the mean and standard error of three cultures in 12-well plates using 10 follicles per well. Asterisks indicate differences between the control group (gray columns) and the vitrified group (black columns); * *p* < 0.05, ** *p* < 0.01.

**Figure 4 animals-13-03563-f004:**
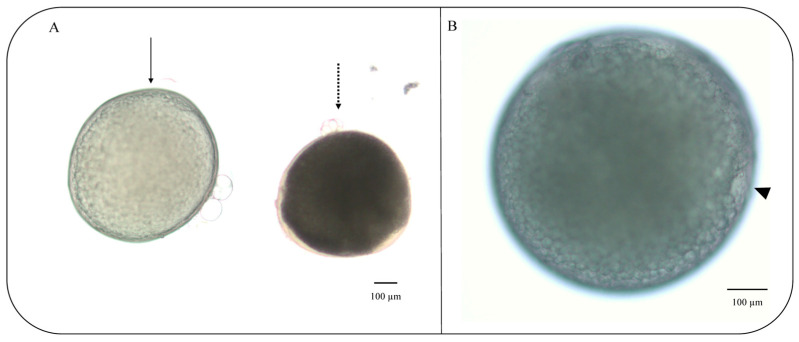
Zebrafish (*Danio rerio*) ovarian follicles cultured in vitro at 27 °C for 270 min: (**A**) matured follicle after culture period from stage III follicle (solid line) and immature follicle that remained in stage III after the culture period (dotted line); (**B**) mature follicle showing micropyle after culture period (triangle). Scale bars indicate 100 μm.

**Figure 5 animals-13-03563-f005:**
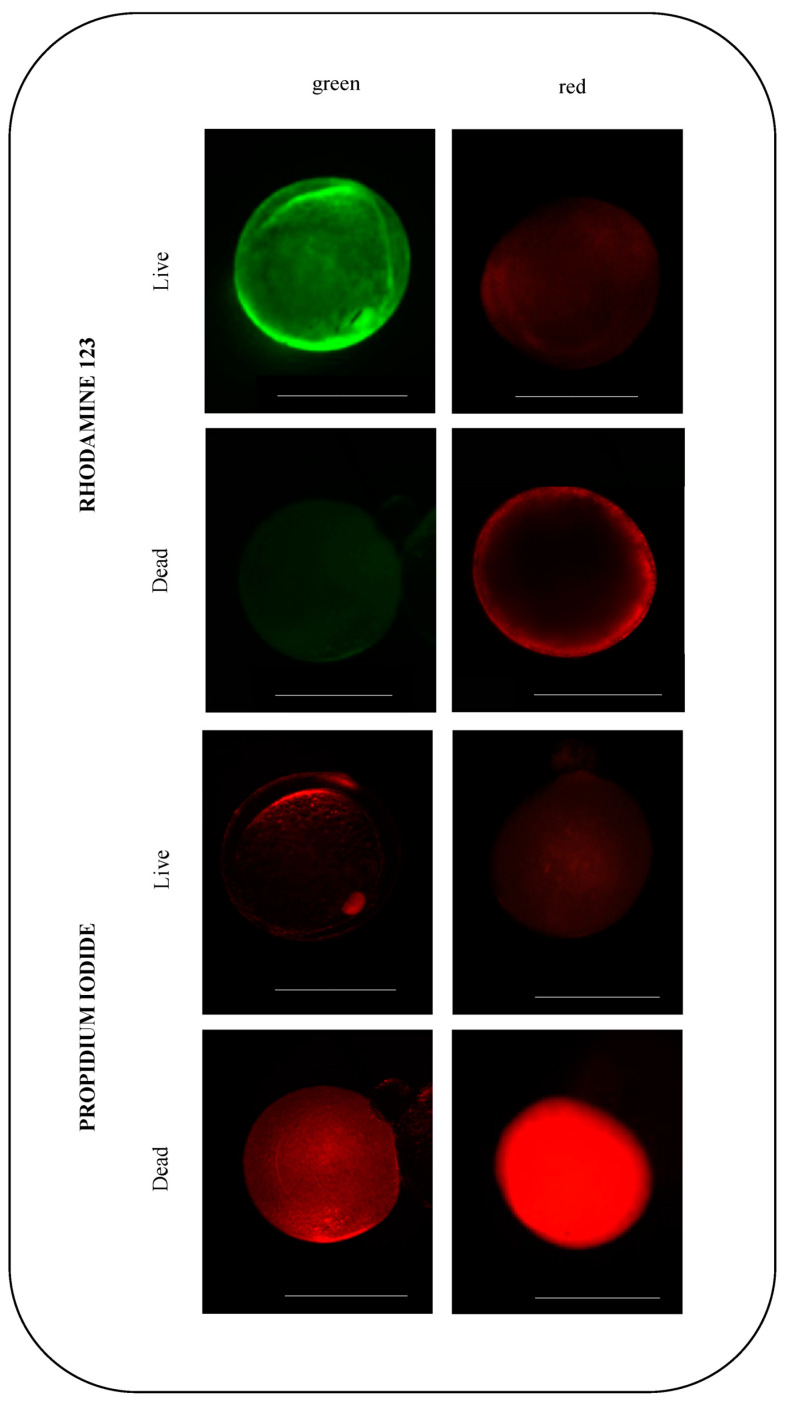
Representative fluorescence images of mature zebrafish (*Danio rerio*) ovarian follicles stained with rhodamine 123 and propidium iodide (PI) for viability assessment after in vitro culture. For each fluorochrome, follicles were divided into living (first row) and dead (second row). Cells stained bright green with rhodamine 123 indicate live cells, and those stained bright red with PI indicate dead cells. Scale bars indicate 0.5 mm.

**Table 1 animals-13-03563-t001:** Primer sequences used for the gene expression analysis in zebrafish ovarian follicles.

Gene	Forward Primer (5′-3′)	Reverse Primer (5′-3′)	Acc Number	Product Length (pb)
Luteinizing hormone/choriogonadotropin receptor (*lhcgr*)	AAGGACGAGTCGCTGAAAC	CACAGCAGAAAGATGTCAGC	AY424302	189
Follicle-stimulating hormone receptor (*fshr*)	GATCCACTCGCTCTTTCGTC	CCGTTCTCGGACACCACTAT	AY424301	169
Cytochrome P450 Family 19 Subfamily A Member 1 (*Cyp19a1*)	TGAGCACGATCTGCTTCAGA	TGCAACTCCTGAGCATCTCT	AF183906	129
Insulin-like growth factor I (*IGF-I*)	CGAAGAAGGTCAAGGTGCTC	GACTGCATCCATCTCTGCAA	AH010825	211
Steroidogenic acute regulatory protein (*Star*)	CCAAGTGCAGATGACCCCAA	GGAAGGTGTGTGCCCTTGTT	NM_131663.1	213
Estrogen receptor 1 (*esr1*)	GGTCCAGTGTGGTGTCCTCT	AGAAAGCTTTGCATCCCTCA	NM_152959.1	204
Estrogen receptor 2a (*esr2a*)	TAGTGGGACTTGGACCGAAC	TTCACACGACCACACTCCAT	NM_180966.2	187
Estrogen receptor 2b (*esr2b*)	TTGTGTTCTCCAGCATGAGC	CCACATATGGGGAAGGAATG	NM_174862.3	156
Caspase3 (*casp3*)	GTGCCAGTCAACAAACAAAG	CATCTCCAACCGCTTAACG	NM_131877.3	172
Tumor protein p53 (*tp53*)	GGCTCTTGCTGGGACATCAT	TGGATGGCTGAGGCTGTTCT	AF365873.1	159
Elongation factor alpha (*ef1-alfa*)	GGTACTACTCTTCTTGATGC	GACTTGACCTCAGTGGTTAC	AY422992	200
Beta-actin	GCCAACACTGTATTGTCTGG	GTACTCCTGCTTGCTAATCC	BC165823	203

## Data Availability

Data are contained within the article and Appendix A.

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
