# Peer review of "Alterations in Gene Expression and the Fatty Acid Profile Impact but Do Not Compromise the In Vitro Maturation of Zebrafish (Danio rerio) Stage III Ovarian Follicles after Cryopreservation"

_animals, 2023, doi:10.3390/ani13223563_

Round 1
Reviewer 1 Report
Comments and Suggestions for Authors
Review report on “Alterations in gene expression and fatty acid profile impact but does not compromise in vitro maturation of zebrafish (Danio rerio) stage III ovarian follicles after cryopreservation”
A brief summary
This manuscript describes the effect of a vitrification, or follicle freezing, technique on three aspects: the expression of certain genes related to reproduction, the fatty acid profile, and the maturation of the follicles from zebrafish. The authors find changes in the expression of most of the genes examined and in some of the fatty acid profiles. The maturation outcomes of the treated follicles could be useful for developing this technique further.
General concept comments
Overall, the methodologies in this manuscript were well executed. However, the graphical figures need more work to enhance them, and the discussion section has some ambiguities to be addressed. The introduction is very short and could be expanded to discuss the genes investigated and the importance of the fatty acid profile to follicle development in fish.
Specific comments
Line 128: Why did you have two reference genes and which one did you use for your calculation?
Line 197: You state that, for the gene expression data, “the results were presented as median +/- standard error”, but in the Figure 1 legend, you state that “The data are present as mean and standard error” so which is it? (also there is a typo in the figure legend – should be “presented” not “present”).
Line 250: Figure 1 legend: Should read “Different letters indicate a difference between the developmental stages within the control group”, as I understand it? Please indicate the sample size in the legend and explain whether each “ball” represents one animal. Italicise the species name.
Figure 1: Please remove the bar/line under the letters a, b or c indicating the significant differences as these distract from the positions of the columns. Please can you make the coloured balls unfilled because they hide the top of the columns and error bars, or alternatively remove the “balls”/filled circles completely as the spread of your data is mainly around the mean/median except in a couple of cases.
Figure 1: Please change the scale on the y-axis of figure 1H to show the bars more clearly – i.e. have the maximum set to 20 minimum.
Figure 2: define PL and TG on the y-axis please. Italicise species name. I don’t see n.s. anywhere on the graphs, although it is in the legend. Please state sample sizes used.
Figure 3: Please put a key showing the control and vitrified groups on the figure. Also, Line 285: this doesn’t seem to be a “Rate” but a percentage of something. Is this the percentage of viable follicles? This is unclear in the legend and graph y-axis label.
Figure 4: Please improve the scale bar clarity. Thankyou.
Figure 5 legend, line 329: Please check grammar and italicise species name. Please improve the scale bars on the images. They are unreadable. Is it propidium iodide or iodate? You have both in the figure/legend.
Is there a graph of the propidium iodide viability data? Which is the trypan blue data? This is not clear – if this is figure 3, please put in the figure legend for clarity.
Discussion
Line 340: What does expressive changes mean? Please reword this sentence.
Line 341-343: In your study, there was no difference in the star gene expression between the different stages in the control group. Please change this sentence – you can’t say that the expression was higher if you haven’t proved it with statistical analysis. Also at line 349: the highest expression wasn’t at that stage – or you have left off the statistical analysis letters from Figure 1E.
Line 347: Do you mean converted to estradiol-17beta?
Line 349-350: Grammar is incorrect. Please also make it clear that you are talking about the present study or another study.
Throughout: be consistent with the naming of the genes e.g. Cyp19a1 or cyp19a1.
Line 354: Did you measure LH or FSH? Then you can only presume that is what is causing the change in Cyp191a expression. You could put in a reference from another paper here to explain this.
Line 357: Is oP450arom the same as cyp19a1? Please clarify in the text.
Line 359: It is not “conversely”, please remove this word. You weren’t talking about vitrification in the previous sentence.
Line 363: Please discuss more about the outcome for the oocytes in the present study of having an altered expression of lhcgr and cyp19a1: will the resulting embryos develop normally? Why has cryopreservation altered their expression, in your view?
Line 406: Please reword “…which can be good”
Line 446: Incorrect grammar.
What about the caspase and TP53 results?
Comments on the Quality of English LanguageThere are some comments on grammar and language in the previous section.
Author Response
Dear reviewer,
I appreciate the points you've highlighted for correction. We've improved the introduction and made the results clearer, in addition to revising the English grammar, by a native and expert professional, as requested. We have accepted all your suggestions, which have already been incorporated into the text, and the questions, point by point, are answered below:
Specific comments
Comments 1: Line 128: Why did you have two reference genes and which one did you
use for your calculation?
Response 1: A coefficient generated from the combination of beta-actin and ef1-alpha geometric means it was used as housekeeping, since it was the most stable found from a set of tested candidate reference genes for gene expression normalization in different stages of follicular development.
Comments 2: Line 197: You state that, for the gene expression data, “the results were
presented as median +/- standard error”, but in the Figure 1 legend, you
state that “The data are present as mean and standard error” so which is
it? (also there is a typo in the figure legend – should be “presented” not
“present”).
Response 2: Corrected accordingly: “The data are presented as mean and standard error”
Comments 3: Line 250: Figure 1 legend: Should read “Different letters indicate a
difference between the developmental stages within the control group”,
as I understand it? Please indicate the sample size in the legend and
explain whether each “ball” represents one animal. Italicize the species
name.
Response 3: Corrected accordingly: “The data are presented as mean and standard error (n=5; each mean is from three biological replicates and duplicate technical replicates). Different letters indicate a difference between the developmental stages within the control group”. Scientific names have been revised to be all in italics
Comments 4: Figure 1: Please remove the bar/line under the letters a, b or c indicating
the significant differences as these distract from the positions of the
columns. Please can you make the colored balls unfilled because they
hide the top of the columns and error bars, or alternatively remove the
“balls”/filled circles completely as the spread of your data is mainly
around the mean/median except in a couple of cases.
Figure 1: Please change the scale on the y-axis of figure 1H to show the
bars more clearly – i.e. have the maximum set to 20 minimum.
Response 4: Corrected accordingly.
Comments 5: Figure 2: define PL and TG on the y-axis please. Italicize species name. I
don’t see n.s. anywhere on the graphs, although it is in the legend.
Please state sample sizes used.
Response 5: Corrected accordingly.
Comments 6: Figure 3: Please put a key showing the control and vitrified groups on the
figure. Also, Line 285: this doesn’t seem to be a “Rate” but a percentage
of something. Is this the percentage of viable follicles? This is unclear in
the legend and graph y-axis label.
Response 6: The legend has been modified to be clearer. In line 285, rate was changed to percentage: “Percentage of zebrafish (Danio rerio) ovarian follicles that developed (mature), remained in stage III (immature), or did not survive (dead) after being cultured for 270 min in vitro at 27 °C. The data shown are the mean and standard error of three cultures in 12-well plates using ten follicles per well. Asterisks indicate differences between the control group (gray columns) and the vitrified group (black columns); * P<0.05, **P<0.01.”
Comments 7: Figure 4: Please improve the scale bar clarity. Thankyou.
Response 7: Corrected accordingly
Comments 8: Figure 5 legend, line 329: Please check grammar and italicize species
name. Please improve the scale bars on the images. They are
unreadable. Is it propidium iodide or iodate? You have both in the
figure/legend.
Response 8: Corrected accordingly; and corrected for propidium iodide, it was a typing error.
Comments 9: Is there a graph of the propidium iodide viability data? Which is the
trypan blue data? This is not clear – if this is figure 3, please put in the
figure legend for clarity.
Response 9: The viability assay with PI was performed qualitatively on a sample of follicles after culture. There is no graph for this, but Figure 5 shows the viable, dead, and immature follicles after culture. Trypan blue staining was used to select the follicles that would be used only before the experiments. Only isolations with more than 95% viable follicles were used in the analysis. Figure 3 shows the data regarding follicle culture: the percentage of follicles that developed in vitro, those that remained in stage III, and the follicles that died. The follicles we used for culture, as well as in other experiments, were viable before culture (they were selected using trypan blue assay), fresh, and vitrified after thawing; thus, we were able to check the physiological viability after culture (using rhodamine and PI).
Discussion
Comments 10: Line 340: What does expressive changes mean? Please reword this
sentence. Line 341-343: In your study, there was no difference in the star gene
expression between the different stages in the control group. Please
change this sentence – you can’t say that the expression was higher if
you haven’t proved it with statistical analysis. Also at line 349: the highest
expression wasn’t at that stage – or you have left off the statistical
analysis letters from Figure 1E.
Response 10: The highest expression of the star gene was expected at the PG stage since this gene provides the substrate for the production of testosterone, which is the initial step for androgen production.
Comments 11: Line 347: Do you mean converted to estradiol-17beta?
Response 11: Yes, Corrected accordingly
Comments 12: Line 349-350: Grammar is incorrect. Please also make it clear that you
are talking about the present study or another study.
Throughout: be consistent with the naming of the genes e.g. Cyp19a1 or
cyp19a1.
Response 12: Corrected accordingly
Comments 13: Line 354: Did you measure LH or FSH? Then you can only presume that
is what is causing the change in Cyp191a expression. You could put in a
reference from another paper here to explain this.
- Response 13: Added reference
Comments 14: Line 357: Is oP450arom the same as cyp19a1? Please clarify in the text.
Response 14: Corrected accordingly; cyp19a1 is the ovarian form of cytochrome P450 aromatase (oP450arom, P450c19a1)
Comments 15: Line 359: It is not “conversely”, please remove this word. You weren’t talking about vitrification in the previous sentence.
Response 15: Corrected accordingly
Comments 16: Line 363: Please discuss more about the outcome for the oocytes in the present study of having an altered expression of lhcgr and cyp19a1: will the resulting embryos develop normally? Why has cryopreservation altered their expression, in your view?
Response 16: we added a discussion to answer your questions: “In the context of in vitro development, the changes observed in the expression of cyp19a1 and lhcgr were not critical. This is because the production and release of E2 would not stimulate the production of vitellogenin by the liver, nor would it inhibit the synthesis of FSH, neither would it stimulate the synthesis of LH in the hypothalamus-pituitary-gonad axis, since the follicle is isolated. In this experimental approach, oocyte maturation was stimulated by the addition of the maturation-inducing hormone (DPH) to the culture medium, which promotes the resumption of meiosis. This can be observed through the breakdown of the germinal vesicle and the emergence of the micropyle. Nevertheless, we cannot estimate whether the embryo generated from a follicle with such changes in gene expression would develop normally, since it would develop without having completed total vitellogenesis.”
Comments 17: Line 406: Please reword “…which can be good”
Response 17: Corrected accordingly
Comments 18: Line 446: Incorrect grammar.
Response 18: Corrected accordingly
Comments 19: What about the caspase and TP53 results?
Response 19: we added a discussion about caspase3 and TP53 results: “The reducing apoptosis markers gene expression, TP53, and Casp3, in PG oocytes after vitrification may be a consequence of the low temperatures in the maintenance of normal tissue function after thawing. Apoptosis is a natural process that helps maintain tissue homeostasis by eliminating damaged or unwanted cells. However, the vitrification process can potentially lead to tissue damage. Reducing apoptosis markers can play a role in promoting cell survival and maintaining tissue integrity, thereby facilitating oocyte restructuration after thawing. In atretic ovaries of Merluccius merluccius, apoptosis genes TP53 and casp3 are upregulated together with autophagocytosis marker genes; and vitellogenesis is the moment at which both apoptosis and autophagy are at their highest expression [48]. Nevertheless, the same higher expression of these markers was not detected in the other stages of follicular development in the control group. In the same way, studies in mammals, including humans, found no differences in apoptosis-related gene expression after vitrification of oocytes and thawing.”
Reviewer 2 Report
Comments and Suggestions for Authors
The manuscript “ALTERATIONS IN GENE EXPRESSION AND FATTY ACID PROFILE IMPACT BUT DOES NOT COMPROMISE in vitro MATURATION OF ZEBRAFISH (Danio rerio) STAGE III OVARIAN FOLLICLES AFTER CRYOPRESERVATION” investigated an interesting topic. I find this study is valuable to this field. Some minor comments should be addressed:
1. Some “in vitro” are not in italic, please keep the format uniform
2. The product length of the PCR process can be added into Table 1.
3. How the primer quality was assessed should be introduced in “2.3 Total RNA extraction and quantitative RT-PCR”
4. In “2.4 Fatty acid profile”, “minutes” can be replaced with “min”
5. Introduce how fatty acid composition are expressed in “2.4 Fatty acid profile”
6. Line 197, the reference gene information can be moved into “2.3 Total RNA extraction and quantitative RT-PCR”
7. It is more accepted that C18:1n9 should be C18:1n-9. If possible, change other fatty acids like this.
8. It may not be necessary to add English name (vaccenic acid) after C18:1n-7. If possible, change other places.
9. Figure 4 is not so clear
10. Othe parts are acceptable.
Author Response
Dear reviewer,
I appreciate your review of our work, and all the points you've highlighted for correction have been accepted and are already incorporated into the text. We conducted a review of English grammar to correct some errors. Point by point are answered below:
Comments 1: Some “in vitro” are not in italic, please keep the format uniform
Response 1: Corrected accordingly
Comments 2: The product length of the PCR process can be added into Table1.
Response 2: Added accordingly
Comments 3: How the primer quality was assessed should be introduced in“2.3 Total RNA extraction and quantitative RT-PCR”
Response 3: We added “Primer efficiencies (in between 85 and 100%) were calculated using serial cDNA dilutions (1:1, 1:4, 1:16, 1:64, and 1:256) from a pool of all the samples, in triplicate, as well as checking for amplification in a negative control without cDNA. The amplification conditions were optimized before the analysis of expression”.
Comments 4: In “2.4 Fatty acid profile”, “minutes” can be replaced with “min”
Response 4: Corrected accordingly
Comments 5: Introduce how fatty acid composition are expressed in “2.4 Fatty acid profile”
Response 5: Added: “The data are presented as means and standard error of the mean.”
Comments 6: Line 197, the reference gene information can be moved into “2.3 Total RNA extraction and quantitative RT-PCR”
Response 6: Corrected accordingly
Comments 7: It is more accepted that C18:1n9 should be C18:1n-9. If possible, change other fatty acids like this.
Response 7: Corrected accordingly
Comments 8: It may not be necessary to add English name (vaccenic acid) after C18:1n-7. If possible, change other places.
Response 8: Corrected accordingly
Comments 9: Figure 4 is not so clear
Response 9: The legend in Figure 4 was rewritten to be clearer: “Zebrafish (Danio rerio) ovarian follicles cultured in vitro at 27 °C for 270 minutes; (A) matured follicle after culture period from stage III follicle (solid line) and immature follicle that remained in stage III after the culture period (dotted line), and (B) mature follicle showing micropyle after culture period (triangle). Scale bars indicate 100 μm.”
Comments 10: Other parts are acceptable.
Reviewer 3 Report
Comments and Suggestions for Authors
Letter to the Authors
animals-2602207-v1
Alterations in gene expression and fatty acid profile impact but does not compromise in vitro maturation of zebrafish (Danio rerio) stage III ovarian follicles after cryopreservation
Fernanda de Mello, Daniel Jaen Alonso, Natalia Pires Vieira Morais de Faria, Victor Hugo Marques, Ethiene Fernandes de Oliveira, Paulo Henrique de Mello, Leandro Godoy, Renata Guimaraes Moreira
230915
Dear authors,
Your potentially interesting MS is good for publishing in the jorunal, if completed. Statement in L409-411 regarding necessity of tests on fertility and embryogenesis yet undone simbolically tells incompleteness of your research. Inseminating the transparent mature eggs with the conspecific sperm and observeing them just a few hours to see the cleavage stage are so easy. I do not understand why you skipped such experiments. Hence first of all, additional experiments on fertilization and embryogenesis of vitrified oocytes are necessary. I am sorry to say, but I recommended the editor to return this MS once to you.
Several additional miscellaneous points should be improved as detailed below.
L2-5
Make the title lowercase or title case to avoid typesetting errors.
L17 simple summary
As the author guide says, it is not a shorter version of the abstract. It works for a news feed for media. When writing it, imagine if you were a media journalist who do not like to read long stories in limited working times. Journalists like to read short stories in a veni-vidi-vici style. When they are interested in your paper seeing your flash presentation, they will read further or call you to gather stories to write.
See examples at:
http://doi.org/10.3390/ani10040633
http://doi.org/10.3390/ani10050778
http://doi.org/10.3390/ani10071140
Consult with a public relation officer of your institution.
L17
preserving -> cryopreserving
L18-21
Successful protocols .. after vitrification. -> delete
L27
to -> in
L29,159,163,474
warming -> thawing
L35
stage III
Make sure which terminology you employ.
L39 keywords
Add a couple of keywords more, which do not appear in the title, to draw attention from wider readership. Hints: storage medium, hypothalamus-pituitary-gonad axis, granulosa cell, oogenesis, ovulation, fertility, etc.
L52-54
[7,6]. However, .. in human oocytes 53 [12]. -> revise
Do not make a simple reference list (A stated this, B analyzed that, C argued it, or alike). Simple reference lists bloat your MS, dilute your originality, and undersell your own research. Use noun phrases to make abstract contents of those references. In your case, particular mammalian species names are not essential.
-> [6,7], whereas vitrification has been successful for oocytes and embryos of several mammals including human [8-12].
L71
fatty acids (grammar) -> fatty acid
L78
follicles; then to perform -> follicles. We observed
Break sentence. "Perform" does not make sense.
L85
maintained ? -> maintained for eight weeks ?
See L133.
L92-93
The follicles were .. described by Selman (1993). (redundant) -> delete
You may cite this reference [24] at the end of next sentence. Citation (#23 and #24) should be swapped (see L136).
L94
analyzed and (verbose) -> delete
the following stages (information deficient) -> the following stages before vitrification
L94-97
primary growth (PG) (stage I .. in diameter)
Make sure which terminology you employ. Introducing more terms, expending more memory spaces of the readers. I think the terminology with PG, PV and VG is better associating with their shortest description. Check thoroughly.
L98,218,230,270,371,384,405,412,421,430,440,457,462,472,492
Make indent. Delete a space above the line.
L98
Developmental stages I, II, and III -> Follicles in PG, PV and VG stages
analyzed for both groups, (does not make sense) -> separated into two groups,
L99
in -> for
maturation -> maturation experiment
fatty acid profile experiments -> fatty acid profile measurements
L100
in both control (fresh) and vitrified groups (redundant) -> delete
L101-102
All procedures were approved .. (Protocol 312/2017). (redundant) -> delete
Or otherwise, omit the institutional review board statement at L518-520.
L108,etc
I suggest you to make abbreviation of the dimethyl sulfoxide as more common "DMSO" rather than "Me2SO".
L110
dimethyl sulfoxide (Me2SO) -> Me2SO (or DMSO, see above)
L115
Bertin Technologies -> Bertin Technologies (Montigny-le-Bretonneux, France)
L119
Applied Biosystems -> Applied Biosystems, Foster City, USA
L121
Thermofisher -> Thermofisher, Waltham, MA USA
L123
Applied Biosystems, Foster City, USA -> Applied Biosystems
L131
table 1 header
GenBank access -> GenBank accession
table 1 body
Make acronyms of gene name stems Italics. Check thoroughly for the main text.
L135
Folch et al. -> the conventional method
A merit of numbered citation is to save readers' short term memory spaces when reading. Readers can go straightforward through the story-flow without outflow of author names [and published years].
L136
phospholipid -> phospholipid (PL)
triglyceride -> triglyceride (TG)
See L227.
L137
Yang's method (1995) ?
Not in the reference list.
L138
Christie's methodology (2003) ?
Not in the reference list.
L139
Scio -> Scion
L149
For the in vitro maturation .. by Seki (2008) was used -> The in vitro maturation of ovarian follicles was examined as descried previously [51]
Citation renumbering is necessary. 51 -> 27 ?
L155
C21H32O3 -> C<sub>21</sub>H<sub>32</sub>O<sub>3</sub>
L174
Trypan Blue ? -> trypan blue ?
See L175.
L178
(see Godoy et al., 2013) -> [14]
L180,184,186,330,332
Rhodamine123 -> rhodamine 123
L191
follicles developmental stage (grammar, does not make sense) -> follicle developmental stages
There were three stages examined.
L202
two cell apoptosis markers (induces misunderstanding) -> two apoptosis markers
L202-203
Fshr and lhcgr .. in the control group -> In the control group, fshr and lhcgr mRNA levels 202 showed differences throughout oogenesis.
Make contrast with L212.
L208-209
Similarly, .. in the control group -> Similarly in the control group, also there were differences in IGF-I and Cyp19a1 mRNA levels
L230
FA (does not make sense) -> delete
L237
PUFA -> polyunsaturated fatty acids (PUFA)
L238
long-chain polyunsaturated fatty acids (LC PUFA) -> long-chain PUFA (LC-PUFA)
See L421,etc.
L250
balls -> dots
present -> presented
L255
Danio rerio -> in Italics
triglycerides -> triglycerides, TG
phospholipids -> phospholipids, PL
You seem presenting spell-outs and their acronyms independently from the main text, and that is OK, but be consistent.
L257
LC PUFA -> LC-PUFA
L261
warmed -> thawed, (insert a comma)
L262
discard (does not make sense) -> detect
L266
the germinal vesicle (GBVD) -> GVBD
See L168.
L298
bars 100 -> bars indicate 100
L329,409
Danio rerio -> in Italics
L330
Propidium Iodide -> PI
See L181.
L335,390,448
Add numbers to the sub-section headers.
L347,355
granulosa -> fillicular
L347-352
In the granulosa cells, .. converting testosterone into E2
Reference(s) is necessary.
L356
In tilapia (Oreochromis niloticus) and medaka (Oryzias latipes), -> delete
L363
observations on medaka and tilapia described above -> previous reports [31,32]
L365-368
Zebrafish with an fsh mutation .. are therefore infertile
This unclear story flow needs revision.
L368-370
In humans, .. post-natal development
This unclear story flow needs revision. What does "fatal" mean?
L391
saturated fatty acids (SFA) -> SFA
See L227.
L400
In vitro studies suggest -> An in vitro study suggested
L406
some metabolic function, which can be good (wordy) -> some {helpful, beneficial} function
L430
oocytes in salmonid species -> fish oocytes (generally speaking)
L434
[44,45,7] -> [7,44,45]
L440-441
Previous studies .. stages of development.
References are necessary.
L440
group, showed -> group showed (delete a comma)
L449
genes expression (grammar) -> gene expression
L451,493,498
warmed -> thawed
L464
stage IV of development -> ovulation ?
L469
body; however, -> body, {though, yet}
L484
dimethyl sulfoxide -> DMSO
See a comment on L108.
L524 references
Check the reference list carefully again from the beginning. Reference lists are frequently hotbeds of errors. You will add, omit or swap citation in the main text on the way revision. Numbering of the references might then shift. If so, readers think you are making irrelevant citation. It is the authors' responsibility that all references are properly cited.
Check thoroughly to make sure:
if paper titles are in lower case (L525,etc),
if scientific names are in Italics (L532,etc),
if journal titles are abbreviated when possible (L530,etc),
if abbreviated journal title words accompany a dot (L543,etc),
if book and journal titles are in Italic title case (L526,etc),
if book editors and publisher information are properly presented (L526,etc),
etc.
See the citation guide at:
https://www.mdpi.com/authors/references/
Several grammatical issues present.
Author Response
Comments 1: Dear authors, your potentially interesting MS is good for publishing in the journal, if completed. Statement in L409-411 regarding necessity of tests on fertility and embryogenesis yet undone symbolically tells incompleteness of your research. Inseminating the transparent mature eggs with the conspecific sperm and observing them just a few hours to see the cleavage stage are so easy. I do not understand why you skipped such experiments.
Hence first of all, additional experiments on fertilization and embryogenesis of vitrified oocytes are necessary. I am sorry to say, but I recommended the editor to return this MS once to you. Several additional miscellaneous points should be improved as detailed below.
Response 1: Dear reviewer,
I appreciate all the points you've highlighted for correction; they have helped make our text clearer and more concise. Following your suggestions, we improved the introduction by including important papers that had not been previously cited. We also conducted a detailed review to include all cited authors. Significant modifications were made to the methodology and results description to make the workflow and results clearer for the reader. The text underwent an extensive English language review by a native English speaker, and we are sure that many misunderstandings have now been clarified. The additional points indicated have all been corrected and are already incorporated into the text. All the changes are in green in the attached file.
About L409-411 I appreciate your thorough review and the concerns you've raised regarding the completeness of our research. I'd like to clarify that our primary focus in this study was to evaluate the post-thawing viability of oocytes. The gamete vitrification technique is well-established for many species of commercial interest and is already a reality. However, the viability of post-thawing oocytes is not always guaranteed. There are few answers to this issue during the thawing process. The experiments we conducted were specifically designed to assess the cryopreservation process and its impact on oocytes' developmental capability.
Inseminating mature eggs with conspecific sperm and monitoring cleavage stages, as you've suggested, are indeed valuable experiments, and we recognize their importance, however, they were not included in our research objectives. We aimed to delve deep into the alterations caused due to the vitrification process and the immediate outcomes upon thawing, focusing on the expression of reproductive pathway genes and fatty acid profile that can impact oocyte survival and in vitro development. We also evaluated the expression of epigenetic regulation genes that will be published soon.
While we understand the significance of the experiments you've mentioned, our research questions and time limited us to a more specialized investigation. We appreciate your feedback and the potential avenues for further research that you've highlighted. In the future, we will consider expanding our studies to include the aspects of fertilization and embryogenesis of vitrified oocytes. Your input is valuable in guiding the direction of our research, and we'll take it into consideration for future investigations.
Comments 2: L2-5
Make the title lowercase or title case to avoid typesetting errors.
Response 2: Corrected accordingly
Comments 3: L17 simple summary
As the author guide says, it is not a shorter version of the abstract. It works for a news feed for media. When writing it, imagine if you were a media journalist who do not like to read long stories in limited working times. Journalists like to read short stories in a veni-vidi-vici style. When they are interested in your paper seeing your flash presentation, they will read further or call you to gather stories to write.
See examples at:
http://doi.org/10.3390/ani10040633
http://doi.org/10.3390/ani10050778
http://doi.org/10.3390/ani10071140
Consult with a public relation officer of your institution.
Response 3: Corrected accordingly, we rewrite the simple summary
Comments 4: L17
preserving -> cryopreserving
Response 4: Corrected accordingly
Comments 5: L18-21
Successful protocols .. after vitrification. -> delete
Response 5: Corrected accordingly
Comments 6: L27
to -> in
Response 6: Corrected accordingly
Comments 7: L29,159,163,474
warming -> thawing
Response 7: Corrected accordingly
Comments 8: L35
stage III
Make sure which terminology you employ.
Response 8: Corrected accordingly
Comments 9: L39 keywords
Add a couple of keywords more, which do not appear in the title, to draw attention from wider readership. Hints: storage medium, hypothalamus-pituitary-gonad axis, granulosa cell, oogenesis, ovulation, fertility, etc.
Response 9: Corrected accordingly
Comments 10: L52-54
[7,6]. However, .. in human oocytes 53 [12]. -> revise
Do not make a simple reference list (A stated this, B analyzed that, Cargued it, or alike). Simple reference lists bloat your MS, dilute your originality, and undersell your own research. Use noun phrases to make abstract contents of those references. In your case, particular mammalian species names are not essential.
-> [6,7], whereas vitrification has been successful for oocytes and embryos of several mammals including human [8-12].
Response 10: The text was rewritten: “Fish oocytes are more challenging to cryopreserve due to their larger size and large quantity of intracellular lipids, which can compromise the success of cryopreservation [6,7]. Nonetheless, vitrification has been successfully carried out in oocytes and embryos (blastocysts and 2-cell-stage) from mice by optimizing the process [8,9], in sheep embryos for embryo transfer programs under extensive conditions and using straws to perform the entire process [10,11], and in human oocytes, thus allowing increased pregnancy rates via artificial reproductive technology [12,13].”
Comments 11: L71
fatty acids (grammar) -> fatty acid
Response 11: Corrected accordingly
Comments 12: L78
follicles; then to perform -> follicles. We observed
Break sentence. "Perform" does not make sense.
Response 12: Corrected accordingly
Comments 13: L85
maintained? -> maintained for eight weeks?
See L133.
Response 13: Corrected accordingly because to the gene expression experiments the females were not maintained at the lab.
Comments 14: L92-93
The follicles were .. described by Selman (1993). (redundant) -> delete
You may cite this reference [24] at the end of next sentence. Citation (#23 and #24) should be swapped (see L136).
Response 14: Corrected accordingly
Comments 15: L94
analyzed and (verbose) -> delete
the following stages (information deficient) -> the following stages before vitrification
Response 15: Corrected accordingly
Comments 16: L94-97
primary growth (PG) (stage I .. in diameter)
Make sure which terminology you employ. Introducing more terms, expending more memory spaces of the readers. I think the terminology with PG, PV and VG is better associating with their shortest description. Check thoroughly.
Response 16: Corrected accordingly
Comments 17: L98,218,230,270,371,384,405,412,421,430,440,457,462,472,492
Make indent. Delete a space above the line.
Response 17: Corrected accordingly
Comments 18: L98
Developmental stages I, II, and III -> Follicles in PG, PV and VG stages
analyzed for both groups, (does not make sense) -> separated into two groups,
Response 18: Corrected accordingly
Comments 19: L99
in -> for
maturation -> maturation experiment
fatty acid profile experiments -> fatty acid profile measurements
Response 19: Corrected accordingly
Comments 20: L100
in both control (fresh) and vitrified groups (redundant) -> delete
Response 20: Corrected accordingly
Comments 21: L101-102
All procedures were approved .. (Protocol 312/2017). (redundant) ->delete
Or otherwise, omit the institutional review board statement at L518-520.
Response 21: Corrected accordingly
Comments 22: L108,etc
I suggest you to make abbreviation of the dimethyl sulfoxide as more common "DMSO" rather than "Me2SO".
Response 22: Corrected accordingly
Comments 23: L110
dimethyl sulfoxide (Me2SO) -> Me2SO (or DMSO, see above)
Response 23: Corrected accordingly
Comments 24: L115
Bertin Technologies -> Bertin Technologies (Montigny-le-Bretonneux, France)
Response 24: Corrected accordingly
Comments 25: L119
Applied Biosystems -> Applied Biosystems, Foster City, USA
Response 25: Corrected accordingly
Comments 26: L121
Thermofisher -> Thermofisher, Waltham, MA USA
Response 26: Corrected accordingly
Comments 27: L123
Applied Biosystems, Foster City, USA -> Applied Biosystems
Response 27: Corrected accordingly
Comments 28: L131
table 1 header
GenBank access -> GenBank accession
table 1 body
Make acronyms of gene name stems Italics. Check thoroughly for the main text.
Response 28: Corrected accordingly
Comments 29: L135
Folch et al. -> the conventional method
A merit of numbered citation is to save readers' short term memory spaces when reading. Readers can go straightforward through the story-flow without outflow of author names [and published years].
Response 29: Corrected accordingly
Comments 30: L136
phospholipid -> phospholipid (PL)
triglyceride -> triglyceride (TG)
See L227.
Response 30: Corrected accordingly
Comments 31: L137
Yang's method (1995) ?
Not in the reference list.
Response 31: Corrected accordingly, we added to the reference list “Yang, Z. Development of a gas chromatographic method for profiling neutral lipids in marine samples. masters, Memorial University of Newfoundland, 1995.”
Comments 32: L138
Christie's methodology (2003) ?
Not in the reference list.
Response 32: Corrected accordingly, we added to the reference list “Christie, W.W., Han, X., In Lipid analysis (fourth edition); 2012, Eds.; Oily Press Lipid Library Series; Woodhead Publishing, p. iv ISBN 978-0-9552512-4-5.
Comments 33: L139
Scio -> Scion
Response 33: Corrected accordingly
Comments 34: L149
For the in vitro maturation .. by Seki (2008) was used -> The in vitro maturation of ovarian follicles was examined as descried previously [51]
Citation renumbering is necessary. 51 -> 27?
Response 34: Corrected accordingly
Comments 35: L155
C21H32O3 -> C<sub>21</sub>H<sub>32</sub>O<sub>3</sub>
Response 35: Corrected accordingly
Comments 36: L174
Trypan Blue? -> trypan blue?
See L175.
Response 36: Corrected accordingly
Comments 37: L178
(see Godoy et al., 2013) -> [14]
Response 37: Corrected accordingly
Comments 38: L180,184,186,330,332
Rhodamine123 -> rhodamine 123
Response 38: Corrected accordingly
Comments 39: L191
follicles developmental stage (grammar, does not make sense) -> follicle developmental stages
There were three stages examined.
Response 39: Corrected accordingly
Comments 40: L202
two cell apoptosis markers (induces misunderstanding) -> two apoptosis markers
Response 40: Corrected accordingly
Comments 41: L202-203
Fshr and lhcgr .. in the control group -> In the control group, fshr and lhcgr mRNA levels 202 showed differences throughout oogenesis.
Make contrast with L212.
Response 41: Corrected accordingly
Comments 42: L208-209
Similarly, .. in the control group -> Similarly in the control group, also there were differences in IGF-I and Cyp19a1 mRNA levels
Response 42: Corrected accordingly
Comments 43: L230
FA (does not make sense) -> delete
Response 43: Corrected accordingly
Comments 44: L237
PUFA -> polyunsaturated fatty acids (PUFA)
Response 44: Corrected accordingly
Comments 45: L238
long-chain polyunsaturated fatty acids (LC PUFA) -> long-chain PUFA(LC-PUFA)
See L421,etc.
Response 45: Corrected accordingly
Comments 46: L250
balls -> dots
present -> presented
Response 46: Corrected accordingly
Comments 47: L255
Danio rerio -> in Italics
triglycerides -> triglycerides, TG
phospholipids -> phospholipids, PL
You seem presenting spell-outs and their acronyms independently from the main text, and that is OK, but be consistent.
Response 47: Corrected accordingly
Comments 48: L257
LC PUFA -> LC-PUFA
Response 48: Corrected accordingly
Comments 49: L261
warmed -> thawed, (insert a comma)
Response 49: Corrected accordingly
Comments 50: L262
discard (does not make sense) -> detect
Response 50: Corrected accordingly
Comments 51: L266
the germinal vesicle (GBVD) -> GVBD
See L168.
Response 51: Corrected accordingly
Comments 52: L298
bars 100 -> bars indicate 100
Response 52: Corrected accordingly
Comments 53: L329,409
Danio rerio -> in Italics
Response 53: Corrected accordingly
Comments 54: L330
Propidium Iodide -> PI
See L181.
Response 54: Corrected accordingly
Comments 55: L335,390,448
Add numbers to the sub-section headers.
Response 55: Corrected accordingly
Comments 56: L347,355
granulosa -> follicular
Response 56: Corrected accordingly
Comments 57: L347-352
In the granulosa cells, .. converting testosterone into E2
Reference(s) is necessary.
Response 57: Corrected accordingly, we added the reference “Nagahama, Y.; Yamashita, M. Regulation of oocyte maturation in fish. Development, Growth & Differentiation 2008, 50, S195–S219, doi:https://doi.org/10.1111/j.1440-169X.2008.01019.x.”
Comments 58: L356
In tilapia (Oreochromis niloticus) and medaka (Oryzias latipes), -> delete
Response 58: Corrected accordingly
Comments 59: L363
observations on medaka and tilapia described above -> previous reports[31,32]
Response 59: Corrected accordingly
Comments 60: L365-368
Zebrafish with an fsh mutation .. are therefore infertile
This unclear story flow needs revision.
Comments 61: L368-370
In humans, .. post-natal development
This unclear story flow needs revision. What does "fatal" mean?
Response 60 and 61: We rewrite the sentences to be clear the story flow: “Reproduction cannot take place without the proper functioning of lhcgr, and it would most likely not be possible to develop viable zebrafish embryos as a result of these gene expression changes [41]. FSH-deficient zebrafish (fshb−/−) have been shown to have their ovary development significantly delayed, although both sexes can be fertile. In contrast, LH-deficient zebrafish (lhb−/−) show normal gonadal growth, but the females fail to spawn [23]. In mice, LH null (lhb−/−) mice are viable, but demonstrate post-natal defects in gonadal growth, defects in folliculogenesis and function, which results in infertility [42].”
Comments 62: L391
saturated fatty acids (SFA) -> SFA
See L227.
Response 62: Corrected accordingly
Comments 63: L400
In vitro studies suggest -> An in vitro study suggested
Response 63: Corrected accordingly
Comments 64: L406
some metabolic function, which can be good (wordy) -> some {helpful, beneficial} function
Response 64: Corrected accordingly
Comments 65: L430
oocytes in salmonid species -> fish oocytes (generally speaking)
Response 65: Corrected accordingly
Comments 66: L434
[44,45,7] -> [7,44,45]
Response 66: Corrected accordingly
Comments 67: L440-441
Previous studies .. stages of development.
References are necessary.
Response 67: Corrected accordingly, we added reference “de Mello, F.; Marques, V.; Pires Vieira Morais de Faria, N.; Godoy, L.; Moreira, R. Vitrification changes the fatty acids profile of zebrafish ovarian follicles at different developmental stages. Cryobiology 2021, 103, 189, doi:10.1016/j.cryobiol.2021.11.109.”
Comments 68: L440
group, showed -> group showed (delete a comma)
Response 68: Corrected accordingly
Comments 69: L449
genes expression (grammar) -> gene expression
Response 69: Corrected accordingly
Comments 70: L451,493,498
warmed -> thawed
Response 70: Corrected accordingly
Comments 71: L464
stage IV of development -> ovulation ?
Response 71: Yes, I changed in the text
Comments 72: L469
body; however, -> body, {though, yet}
Response 72: Corrected accordingly
Comments 73: L484
dimethyl sulfoxide -> DMSO
See a comment on L108.
Response 73: Corrected accordingly
Comments 74: L524 references
Check the reference list carefully again from the beginning. Reference lists are frequently hotbeds of errors. You will add, omit or swap citation in the main text on the way revision. Numbering of the references might then shift. If so, readers think you are making irrelevant citation. It is the authors' responsibility that all references are properly cited.
Check thoroughly to make sure:
if paper titles are in lower case (L525,etc),
if scientific names are in Italics (L532,etc),
if journal titles are abbreviated when possible (L530,etc),
if abbreviated journal title words accompany a dot (L543,etc),
if book and journal titles are in Italic title case (L526,etc),
if book editors and publisher information are properly presented(L526,etc),
etc.
See the citation guide at:
https://www.mdpi.com/authors/references/
Response 74: We used the Zotero editor to make sure that no reference was left out of the list and that it was cited correctly
Round 2
Reviewer 3 Report
Comments and Suggestions for Authors
Letter to authors
animals-2602207-v2
Alterations in gene expression and fatty acid profile impact but do not compromise in vitro maturation of zebrafish (Danio rerio) stage III ovarian follicles after cryopreservation
Fernanda de Mello, Daniel Jaen Alonso, Natalia Pires Vieira Morais de Faria, Victor Hugo Marques, Ethiene Fernandes de Oliveira, Paulo Henrique de Mello, Leandro Godoy, Renata Guimaraes Moreira
231023
Dear authors,
I am sorry to say I recommended the editor to return your MS. You allege you assessed if the developmental capacity was compromised (L89), but you did not. The assessment could simply and clearly be done by observing fertilization and embryogenesis of vitrified oocytes.
This MS is unsuitable for publication in the journal, because of the lack of a simplest experiment on developmental capacity of the eggs.
I am sorry for this review result.
Author Response
Dear Reviewer,
Thank you for your detailed review of our manuscript. We appreciate the time and effort you have dedicated to evaluating our work. We would like to address your concerns and clarify once again the main objective of our study.
1.
Objective Clarification:
The primary aim of our research was to evaluate the in vitro growth and maturation capacity of immature oocytes post thawing.
2.
Assessment of Developmental Capacity (L89):
We acknowledge your comment regarding our statement in line 89 about assessing developmental capacity. We understand the importance of clarity and precision in scientific communication. Perhaps, it was not clear that our objective was to evaluate the oocyte developmental capacity only in vitro, since we work with immature oocytes, that is, not capable of fertilization.
3.
Methodological Approach:
Our focus was on the in vitro growth and maturation capacity; our experimental design was intentionally centered on post-thaw development in vitro, and we did not specifically investigate fertilization and embryogenesis parameters.
4.
Rationale for Experiment Design:
The decision to focus only on in vitro growth and maturation was motivated by the need to explore this specific aspect of oocyte viability post-thawing in a more controlled way. Gametes cryopreservation is a well-established technique for some fish aquaculture species; nevertheless, little is known about their post-thawing impact on the cells. What is happening at the molecular level during oocyte development can impede growth and maturation? Many parameters were analyzed throughout this work. Currently, gamete cryopreservation of fish is applied in genetic conservation in germplasm banks, but in aquaculture, only semen cryopreservation is used routinely on farms. We believed that our study focused on the capacity for in vitro growth and maturation post-thawing contributed valuable information on oocyte genetic and physiological viability.
5.
Consideration of Feedback:
We believe that vitrification is an important tool for the production of aquaculture species, enabling the availability of gametes throughout the year, regardless of the reproductive period or maintenance of the animals, among other advantages. The results presented by us are valuable regarding the use of these post-thawing oocytes and therefore we believe that our work fits into the special edition of this journal to which this work was submitted.
Sincerely,
Dra. Fernanda de Mello
University of São Paulo
fernandade. mello@gmail.com